# PTHrP Modulates the Proliferation and Osteogenic Differentiation of Craniofacial Fibrous Dysplasia-Derived BMSCs

**DOI:** 10.3390/ijms24087616

**Published:** 2023-04-20

**Authors:** Lihang Shen, Yang He, Shuo Chen, Linhai He, Yi Zhang

**Affiliations:** 1Department of Oral and Maxillofacial Surgery, Peking University School and Hospital of Stomatology, Beijing 100081, China; chercherh@126.com (L.S.);; 2First Clinical Division, Peking University School and Hospital of Stomatology, Beijing 100034, China

**Keywords:** fibrous dysplasia, BMSCs, PTHrP, cAMP/PKA/CREB, Wnt/β-catenin

## Abstract

Fibrous dysplasia (FD) is a skeletal stem cell disease caused by mutations in the guanine nucleotide-binding protein, alpha-stimulating activity polypeptide (*GNAS*) gene, which results in the abnormal accumulation of cyclic adenosine monophosphate (cAMP) and hyperactivation of downstream signaling pathways. Parathyroid hormone-related protein (PTHrP) is secreted by the osteoblast lineage and is involved in various physiological and pathological activities of bone. However, the association between the abnormal expression of PTHrP and FD, as well as its underlying mechanism, remains unclear. In this study, we discovered that FD patient-derived bone marrow stromal cells (FD BMSCs) expressed significantly higher levels of PTHrP during osteogenic differentiation and exhibited greater proliferation capacity but impaired osteogenic ability compared to normal control patient-derived BMSCs (NC BMSCs). Continuous exogenous PTHrP exposure on the NC BMSCs promoted the FD phenotype in both in vitro and in vivo experiments. Through the PTHrP/cAMP/PKA axis, PTHrP could partially influence the proliferation and osteogenesis capacity of FD BMSCs via the overactivation of the Wnt/β-Catenin signaling pathway. Furthermore, PTHrP not only directly modulated cAMP/PKA/CREB transduction but was also demonstrated as a transcriptional target of CREB. This study provides novel insight into the possible pathogenesis involved in the FD phenotype and enhances the understanding of its molecular signaling pathways, offering theoretical evidence for the feasibility of potential therapeutic targets for FD.

## 1. Introduction

Fibrous dysplasia (FD) is a non-hereditary benign bone disease characterized by the replacement of bone and marrow by abnormal fibrous bone tissues. When FD occurs in craniofacial bones, its clinical manifestations primarily include facial deformities, malocclusion, visual impairment, and dysaudia [1]. If combined with café-au-lait spots and hyperfunctioning endocrinopathies, it is termed McCune–Albright syndrome (MAS) [2]. The main pathogenesis of FD/MAS is considered to be missense mutations in the *GNAS* gene, which encodes the α-subunit of stimulatory G-protein (G_s_α). Most mutations occur at Arg201, including Arg-to-His (R201H) and Arg-to-Cys (R201C) [3]. These mutations interfere with the intrinsic GTPase activity of G_s_α in FD-derived bone marrow stromal cells (BMSCs), resulting in continuous constitutive receptor activation and the triggering of inappropriate cAMP/PKA/CREB signaling pathway transduction [4,5]. Past studies have also suggested other possible etiologies of FD, including the overexpression of HDAC8 [6], overactivation of the Wnt/β-catenin signaling pathway [7,8], unbalanced alteration of the OPG/RANKL ratio [9], and higher secretion of IL-6 [10,11] and fibroblast growth factor-23 [12,13]. All the above-mentioned studies could be summarized as the impaired osteogenic potential of osteoblasts or increased osteoclasts formation.

Interestingly, there is a mosaic phenomenon in FD lesions; that is to say, the mutated BMSCs coexist with wild-type BMSCs [11,14]. The non-mutated BMSCs seem to retain normal cell function; however, the mutated BMSCs could result in osteogenic defects but only be achieved under the condition of coexistence with wild-type cells [15,16]. Figuring out the interactions of the two genotypes of the cells may contribute to further understanding of the progression of FD. Therefore, it might be necessary to explore the specific contributions of mutated BMSCs to osteogenic differentiation disorder and elucidate the mechanism by which mutated cells influence the behavior of wild-type cells in the coexistent system of FD.

Parathyroid hormone-related protein (PTHrP) is secreted by the osteoblast lineage and functions as a ligand of the parathyroid hormone 1 receptor (PTH1R) through the autocrine/paracrine pathway, participating in bone metabolism [17]. Acting as a physiological regulator of bone turnover, PTHrP stands on the lever of osteogenesis versus osteoclastogenesis [18,19,20], depending on the amount and dosing frequency of PTHrP. Previous studies have reported that MAS-derived osteoblasts produced higher PTHrP levels than the healthy control and other bone diseases [21,22,23]. It suggests a possible relationship between PTHrP and the occurrence or development of FD.

This study demonstrates the mechanisms of PTHrP on the overactivation of cAMP/PKA/CREB and canonical Wnt/β-Catenin signaling pathways, which might disrupt the osteogenic lineage differentiation to mature osteoblasts and influence cell proliferation capacity. Additionally, therapeutic intervention targeting PTHrP might offer an alternative treatment for FD.

## 2. Results

### 2.1. Characteristics of FD Lesion-Derived BMSCs

The patients’ clinical information and *GNAS* mutation analysis are shown in Appendix A. In the FD lesions, the histological results displayed distorted trabecular bone with hypomineralization and massive fibrous-like tissues replacing normal bone tissues. The early osteogenic marker, RUNX2, was significantly upregulated in the osteoblasts and fibroblast-like cells. The late osteogenic marker, OCN, was only occasionally located in and beside trabecular bones, implying the mineralization deficiency of the FD bones (Figure 1A). The IHC and IF results revealed that PTHrP, as well as active β-Catenin, were abundantly distributed in FD-derived lesion tissues and BMSCs (Figure 1A,B and Appendix A).

In vitro, a significantly stronger proliferation ability of FD BMSCs than normal control (NC) BMSCs was observed via a BrdU incorporation assay and CCK8 assay (Figure 1B,C). Compared to the NC BMSCs, significantly higher intracellular cAMP levels and secreted PTHrP expression levels could be detected in the FD BMSCs by an ELISA (Figure 1D,E). The osteogenic potential of the NC and FD BMSCs was also compared, and the results indicated that FD BMSCs tended to be immature, with significantly higher sustained PTHrP (Appendix A) and RUNX2 expression, but lower OCN expression and less alizarin red staining, even after 14 days of osteoinductive differentiation (Figure 1F,G). Furthermore, significantly higher expression of canonical Wnt/β-Catenin pathway-related proteins (including active β-Catenin, p-GSK-3β, and TCF1/TCF7) and the phosphorylation level of p-CREB also implied their possible participation in the impaired osteogenic differentiation process of FD BMSCs (Figure 1B,F).

To identify the osteogenic potential in vivo, the NC BMSCs and FD BMSCs were incubated with Bio-Oss^®^ material and transplanted into each side of 6-week-old SCID beige mice’s backs subcutaneously. The specimens were harvested after 8 weeks of culture, and histological studies were performed (Figure 1H). The morphological results showed that the NC BMSCs-transplanted group presented more newly formed bone and mature osteoblasts compared with the FD BMSCs-transplanted group (Figure 1I).

Due to prominent PTHrP overexpression in the FD BMSCs, it is reasonable to suspect that PTHrP is involved in the endogenous pathogenesis of craniofacial FD.

### 2.2. PTHrP Could Modulate the cAMP/PKA/CREB and Canonical Wnt/β-Catenin Signaling Pathways Simultaneously, Thereby Influencing the Osteogenic Differentiation and Proliferation Capacity of BMSCs

Sustained administration of exogenous PTHrP to NC BMSCs notably upregulated the phosphorylation levels of p-PKA and p-CREB and the synthesis of cAMP, indicating activation of the cAMP/PKA/CREB signaling pathway (Figure 2A,D). In response to PTHrP treatment, proteins related to the canonical Wnt/β-Catenin signaling pathway were also significantly upregulated (Figure 2A and Appendix A). Moreover, an enhanced proliferation capacity of the PTHrP-treated NC BMSCs was observed using BrdU and CCK8 assays (Figure 2B,C). The PTHrP then led to an increase in the early osteogenic differentiation marker, RUNX2, while decreasing the late osteogenic differentiation markers, OCN, demonstrating the inhibition effect of PTHrP on the osteogenic differentiation of the NC BMSCs (Figure 2E,F). In vivo, the addition of exogenous PTHrP attenuated the formation of mineralized tissues in the NC BMSC-transplanted group (Figure 2G and Appendix A).

Three siRNA sequences were employed to knock down PTHrP expression levels in FD BMSCs; si-*PTHLH*-RNA1 and si-*PTHLH*-RNA3 more effectively inhibited both intracellular and extracellular expression of PTHrP (Figure 2H and Appendix A). The reductions in cAMP (Appendix A) and the decrease in phosphorylation levels of PKA and CREB (Figure 2I) confirmed the direct modulation of PTHrP on the cAMP/PKA/CREB signaling pathway. In addition to β-Catenin, both p-GSK-3β, and TCF1/TCF7 were downregulated by siRNA, providing further evidence of the regulatory role of PTHrP in the canonical Wnt/β-Catenin signaling pathway (Figure 2I and Appendix A). RUNX2, previously demonstrated as a direct target of the canonical Wnt/β-Catenin signaling pathway [24], was also downregulated following PTHrP knockdown. Concurrently, the expression of OCN and the mineralization degree of FD BMSCs were significantly rescued (Figure 2I,K). The BrdU results indicated that the reduction of PTHrP could decrease the proliferation capacity of FD BMSCs (Figure 2J).

Furthermore, we utilized 6-TG, reported as an effective PTHrP inhibitor [25,26], to inhibit the effects of PTHrP on FD BMSCs (Figure 2L and Appendix A). It came out that 6-TG significantly suppressed PTHrP expression and decreased the cAMP/PKA/CREB and Wnt/β-Catenin signaling pathway-related proteins (Figure 2L and Appendix A). The utilization of 6-TG also reversed the excessive cell proliferation capacity of the FD BMSCs (Figure 2M and Appendix A). After 14 days of induction of osteogenic differentiation, similar results as those of siRNA were observed (Figure 2N,O). In vivo, 6-TG also improved the osteogenic phenotype of FD BMSCs to some extent (Figure 2P and Appendix A).

These findings suggested that PTHrP could modulate the cAMP/PKA/CREB and Wnt/β-Catenin signaling pathways simultaneously. Furthermore, excessive PTHrP production by FD BMSCs could be responsible for diminished osteogenic differentiation and the elevation of proliferative capacity.

### 2.3. PTHrP/cAMP/PKA Axis Could Influence the Osteogenesis and Proliferation Capacity of FD BMSCs via the Canonical Wnt/β-Catenin Signaling Pathway

Upon binding with the PTH1R, PTHrP promotes the coupling and activation of heterotrimeric G proteins. Subsequently, G_s_α activates adenylate cyclases, resulting in cAMP synthesis and PKA activation [27]. Based on previous studies, PKA might be the primary mediator connecting PTHrP and the canonical Wnt/β-Catenin signaling pathway in FD BMSCs. H89, a PKA inhibitor, was employed to suppress the effect of PKA on the FD BMSCs. Consequently, the expression intensity of p-GSK-3β and active β-Catenin decreased with increasing H89 concentrations (Figure 3A,B). These findings demonstrated the regulatory role of the PTHrP/cAMP/PKA axis on the Wnt/β-Catenin signaling pathway.

According to previous studies regarding the involvement of the Wnt/β-Catenin signaling pathway in bone mass regulation and cell proliferation [28,29], we hypothesized that the effect of PTHrP on FD BMSCs might be partially mediated through the canonical Wnt/β-Catenin signaling pathway. In this study, FD BMSCs were treated with XAV-939, a typical β-Catenin inhibitor. The WB results revealed a significant decrease in active β-Catenin, TCF1/TCF7, and RUNX2 expression (Figure 3C). In response to the reduced activation of the Wnt/β-Catenin signaling pathway, BrdU and CCK8 assays displayed an obvious downward trend in the proliferation capacity of FD BMSCs (Figure 3D,E). Conversely, the proliferation capacity of NC BMSCs could be significantly upregulated by exogenous Wnt3a (100 ng/mL) (Figure 3F,G). These findings suggested that the effects of PTHrP on the cell proliferation capacity of FD BMSCs could be achieved through the canonical Wnt/β-Catenin signaling pathway.

At day 14, following osteogenic induction, XAV-939 significantly decreased RUNX2 expression; however, it failed to restore the mineralization degree of the FD BMSCs (Figure 3H,I). Nonetheless, exogenous Wnt3a could reverse the effects of 6-TG on the RUNX2 and OCN expression in FD BMSCs, while also influencing active β-Catenin and the phosphorylation levels of CREB and GSK-3β (Figure 3J,K and Appendix A). Subsequently, a continuous exogenous PTHrP was added to NC BMSCs throughout the osteogenic differentiation process. It turned out that 6-TG could still significantly rescue the effects of exogenous PTHrP on the canonical Wnt/β-Catenin signaling pathway and osteogenesis-related markers (Figure 3L,M). These results implied that PTHrP could partly modulate the osteogenesis of FD BMSCs through the canonical Wnt/β-Catenin signaling pathway.

### 2.4. Positive Feedback Regulation Might Exist between the cAMP/PKA/CREB Signaling Pathway and PTHrP

An excessive accumulation of cAMP was considered a major physiological characteristic that would ignite a series of aberrant biological activities in FD BMSCs. Consequently, we treated NC BMSCs with Dibutyryl-cAMP (db-cAMP), a stabilized cAMP analog and selective PKA activator, to simulate the physiological processes of FD [6,11]. As a result, the db-cAMP-treated NC BMSCs exhibited upregulated expression of active β-Catenin, RUNX2, and the phosphorylation levels of PKA, CREB, and GSK-3β, but downregulated expression of OCN and mineralization degree (Figure 4A,B). Interestingly, alterations in PTHrP expression were also observed with increasing db-cAMP and H89 concentrations (Figure 3A and Figure 4A), suggesting a bidirectional regulatory relationship between PTHrP and the cAMP/PKA/CREB signaling pathway. When treated with KG-501 (25 μM), an inhibitor of CREB target gene transcription, noticeable reductions in PTHrP expression were detected in both the db-cAMP-treated NC BMSCs and FD BMSCs (Figure 4C,D). We searched the JASPAR database for the putative transcription factors of *PTHLH* and identified two putative binding sites with CREB. Significant enrichment of CREB was detected by ChIP-qPCR at two putative binding sites in the PTHrP promoter region in FD BMSCs. When treated with KG-501 (25 μM), ChIP enrichment efficiencies decreased significantly (Figure 4E). Dual-Luciferase reporter assay was implemented to further verify the transcriptional regulation of CREB on the *PTHLH* promoter of FD BMSCs and 293T cells, demonstrating that mutations of the two putative binding sites effectively reduced *PTHLH* promoter reporter activity (Figure 4F–H). Collectively, these findings suggested that CREB could promote *PTHLH* transcription by directly binding to its promoter region in FD BMSCs.

Based on these results, we concluded that an inappropriate production of PTHrP in FD BMSCs appears to be attributed to the positive feedback regulation of the cAMP/PKA/CREB signaling pathway.

## 3. Discussion

Throughout the process of osteogenic differentiation, PTHrP is mainly produced by immature osteoblasts and works on mature osteoblasts and osteocytes by paracrine secretion [30,31]. In Appendix A, a fluctuating trend in PTHrP expression was observed during the differentiation process of the NC BMSCs. Previous experimental results suggest that PTHrP intracellular retention would increase initially during the osteogenic differentiation of MCSs, followed by a decrease after the first seven days [32]. In the late phase of the osteogenic differentiation, the amount of extracellular secretion of PTHrP was significantly higher than that of intracellular retention. These might explain why the intracellular level of PTHrP in the NC BMSCs was extremely low but exhibited a high secreted level at 14 days of osteogenic differentiation. In comparison, the FD BMSCs were characterized by a sustained high-level production of PTHrP intracellularly and extracellularly throughout the osteogenic differentiation process.

The activated canonical Wnt/β-Catenin signaling pathway is responsible for regulating bone mass [33,34]; however, it will restrain osteogenic maturation when overactivated [35,36,37,38]. In this study, significant activation of the Wnt/β-Catenin signaling pathway was detected, as previously described [7,8]. PKA was demonstrated to be capable of inactivating GSK-3β by phosphorylating the amino acid Ser9 [39], which could facilitate the accumulation and subsequent nuclear translocation of β-Catenin. Our findings provided strong evidence that the overexpression of PTHrP in FD BMSCs was capable of stimulating more activated PKA, promoting the phosphorylation of GSK-3β; this eventually resulted in the excessive activation of the canonical Wnt/β-Catenin signaling pathway.

In this study, XAV-939 was utilized to inhibit the effects of the Wnt/β-Catenin signaling pathway on FD BMSCs. It effectively reduced RUNX2 expression, though its impact on terminal osteogenic differentiation markers was less pronounced. This might be attributed to the complex cross-talks among osteogenesis-related signaling pathways. Furthermore, previous studies have indicated that FD BMSCs exhibit higher OCN and OPN mRNA expression, as well as an enhanced osteogenic phenotype when β-Catenin is knocked down or treated with LGK-974 [7,8]. Moreover, exogenous Wnt3a counteracted the effects of 6-TG on the FD BMSCs, providing evidence that the influence of PTHrP on osteogenesis could be partially achieved through the canonical Wnt/β-Catenin signaling pathway. Nevertheless, given the significant role of the Wnt/β-Catenin signaling pathway in fundamental cellular activities, additional evidence is required to support the feasibility of the therapeutic targeting of the Wnt/β-Catenin signaling pathway.

Furthermore, PTHrP was demonstrated to enhance the proliferative capacity of osteoblasts [40,41]. A substantial increase in proliferative potential was observed in NC BMSCs when treated with PTHrP or Wnt3a. Conversely, both 6-TG and XAV-939 could reduce the proliferation capacity of FD BMSCs considerably. Due to the crucial function of the Wnt/β-Catenin signaling pathway in regulating cell proliferation [29,42] and the modulatory effect of PTHrP on β-Catenin, it can be inferred that the influence of PTHrP on proliferation could also be mediated via the canonical Wnt/β-Catenin signaling pathway. Additionally, PTHrP was reported to stimulate rat bone marrow cell proliferation through protein kinase C activation of the Ras/mitogen-activated protein kinase signaling pathway [43]. Furthermore, the cAMP/PKA/CREB signaling pathway was also reported to be closely associated with cell growth, proliferation, differentiation, and cycle regulation [44], which explained the cause of the stronger proliferative ability of FD BMSCs, from another perspective.

Nearly all of the reported transgenic mouse models for FD exhibited expansile bone deformities, poorly mineralized trabeculae with dense fibrous matrices, and diminished bone marrow spaces [8,45,46,47,48]. Nevertheless, the local transplantation of BMSC-attached scaffolds was also widely acknowledged as an FD research model [6,16,49]. In this study, we employed the latter method and obtained impaired osteogenic phenotypes. Continuous high-dose PTHrP injection would inhibit osteogenesis and promote osteoclast formation [18,19,50,51]. The continuous exogenous application of PTHrP in the NC group led to mineralization disorders, while 6-TG appeared to counteract the effects of PTHrP in the FD group to some extent. However, a local transplantation approach lacks the participation of osteoclasts in bone remodeling and fails to represent the comprehensive condition of FD. Although 6-TG has been reported as a specific PTHrP inhibitor targeting PTHrP promoters [25,26,52], the adverse effects of 6-TG-like hepatotoxicity and myelosuppression are still non-negligible. Consequently, the dosage and frequency of this treatment necessitate further investigation.

The PTH1R is a class B G-protein-coupled receptor that primarily couples to the G_s_ protein. Through a paracrine mechanism, immature osteoblast-derived PTHrP specifically binds to the PTH1R of peripheral osteoblast lineages, participating in the modulation of calcium homeostasis and bone turnover [17,53,54]. In this study, ChIP and dual luciferase reporter assay results directly confirmed the transcriptional activity of CREB on the *PTHLH* promoter. Concurrently, both siRNA-*PTHLH* and 6-TG reduced intracellular cAMP levels and the phosphorylation levels of PKA and CREB. Collectively, the constitutively activated mutations of the G_s_α protein in FD BMSCs could enhance the expression level of PTHrP. Conversely, through an autocrine pathway, PTHrP might regulate its own activation of the cAMP/PKA/CREB signaling pathway by binding to the PTH1R. Due to mosaic mutations, despite the limited mutation rate of BMSCs in the FD lesions [55,56], the same spatiotemporal distribution enables the interactive paracrine effect between mutation-bearing cells and wild-type cells [16,57]. It seems reasonable that excessive PTHrP from mutated cells not only acts on itself but also binds to non-mutated cells, resulting in impaired osteogenic differentiation and maturation disorders. Previous studies have reported that the number of mutant BMSCs and disease burden will decrease with age [55,56]. Interestingly, PTHrP production also seems to decline with aging [58]. However, the direct causality still needs more clinical evidence.

In addition to local malformations, bone FD could also be accompanied by hyperfunctioning endocrinopathies [57,59], such as gonadotropin-independent gonadal function, nonautoimmune hyperthyroidism, growth hormone excess, and neonatal hypercortisolism. According to a meta-analysis, the incidence of *GNAS* mutations in FD was 86% (264/307) [3]. Besides gene mutation and PTHrP, hormone levels might also be exacerbating factors that promote the accumulation of intracellular cAMP for FD BMSCs systemically [60].

PTHrP was initially identified as an etiological factor of malignancy-induced hypercalcemia [61]. Subsequently, researchers demonstrated that excessive expression of PTHrP from osteoblasts could promote local upregulated osteoclastic activity, contributing to tumor bone metastasis [62,63]. In conjunction with the histological characteristics of clinical FD tissue samples [9,11] and transgenic mouse models [47,48], which both reported overloaded osteoclastogenesis, it is imperative to determine whether PTHrP is associated with excessive osteoclast activity in FD lesions. Recently, medications targeting osteoclasts, such as bisphosphonates and denosumab, have been applied to animal models [48,64] and clinical trials [65,66,67,68], achieving positive results to some extent. We are eager to further investigate the underlying relationship between PTHrP and osteoclasts in FD lesions. Moreover, considering the challenges of gene therapy targeting *GNAS* and the effectiveness of inhibiting PTHrP in this study, therapeutics targeting PTHrP offer significant clinical value for future research.

Based on radiological characteristics, craniofacial FD lesions typically present with a ground-glass appearance with or without cystic changes (typical CT images and H&E staining are shown in Appendix A). Generally, relatively low-density areas on CT scans primarily consist of fibrous-like tissues, while ground-glass lesions usually contain more trabecular bone components. It is uncertain whether the coexisting lesions exert different physiological effects and rates of disease progression. Furthermore, whether surgical or medical therapeutic strategies should be determined by the proportion of fibrous ingredients may be worth further research and discussion.

Limited by the number of cases included in this study, further investigation is required to expand the sample size and validate the generalizability of our findings. Additionally, employing high-throughput sequencing techniques to compare cases with other mutation types may facilitate a more comprehensive analysis of the impact of genetic mutation types on potential pathogenic genes, proteins, and signaling pathways within cells, thereby informing clinical pharmacotherapy. Although the BMSCs used in this study can, to some extent, reflect the characteristics of mosaic mutations in FD, the presence of wild-type cells in the primary cell population may still interfere with the study of mutant cells. Isolated FD BMSCs could potentially be engineered using CRISPR-Cas9 technology to construct *GNAS* mutant BMSCs for further investigation of their features. Moreover, clinical observations indicate that some patients exhibit slow progression while others display rapid advancement of lesions within a relatively short period, suggesting that there may be distinct physiological and pathological changes during disease progression and quiescent phases. Biomarkers indicative of these significant changes may provide guidance for disease staging, surgical interventions, and pharmacological treatments. Considering that PTHrP is involved in both bone formation and resorption, this dual-regulatory factor warrants increased attention in future FD research. Our study highlights the potential of PTHrP as a therapeutic target for craniofacial fibrous dysplasia and provides a theoretical basis for the feasibility of disease treatment. However, further research is necessary to overcome the aforementioned limitations and offer more robust support for the diagnosis and treatment of FD patients.

## 4. Materials and Methods

### 4.1. Primary Cell Acquisition and Culture

The Ethics Committee of the Peking University School and Hospital of Stomatology approved this study (PKUSSIRB-202278106). FD BMSCs were isolated from FD patients’ lesions. NC BMSCs were derived from excess bone tissues of patients who underwent orthognathic surgery and had no metabolic bone diseases. In this study, the age, gender, sampling site, mutation sequencing, and computed tomography (CT) scans of three healthy controls and three FD patients are shown in Appendix A. The average age of the FD patients was 20.33 ± 2.08 years, with a male-to-female ratio of 2:1, while the average age of the NC group was 22.67 ± 2.52 years, with a male-to-female ratio of 2:1. The FD patients and healthy controls included in the study were all in good overall health, with no endocrine disorders, bone metabolic diseases, or café-au-lait skin spots. Apart from the craniofacial FD symptoms, no apparent symptoms of FD lesions in the trunk or limb bones were observed. Informed consent was obtained from volunteers enrolled in this study. Results of BMSCs identification by flow cytometry analysis (BD Human MSC Analysis Kit, BD, NJ, USA) are shown in Appendix A. Primary BMSCs were cultured in a medium composed of α-MEM (Gibco, CA, USA), 10% FBS (Sigma-Aldrich, MO, USA), 100 U/mL antibiotics–antimycotics (Gibco, CA, USA), and 2 mM L-glutamine (Gibco, CA, USA) and incubated at 37 °C and 5% CO_2_. The medium was changed every three days. Only 3–5 passages were used throughout the experiment.

### 4.2. CCK8 Colorimetric Assay and BrdU Incorporation Assay

In this study, 2000 cells per well were seeded onto 96-well plates and incubated for the first 24 h. Subsequently, the supernatants were removed and replaced by 100 μL of a fresh, complete medium with 10 μL of CCK8 solution (Dojindo, Kumamoto, Japan). After incubation for 2 h at 37 °C in the dark, the absorbance was measured at 450 nm using a microplate reader. The entire process lasted for 8 days.

Another 50,000 cells per well were plated onto 12-well plates, and 10 μM BrdU (Sigma-Aldrich, MO, USA) was added to each well with a fresh, complete medium when the cells were at approximately 60–70% confluence. After incubation for 12 h, the cells were fixed with 4% paraformaldehyde and subjected to the subsequent immunofluorescence processes in 4.5.

### 4.3. Animal Model

All animal procedures were performed according to the guidelines of the Animal Care and Use Committee of Peking University (PUIRB-LA2022623). Approximately 5 × 10^6^ BMSCs were co-incubated with Bio-Oss^®^ (0.5 g, Granules 0.25–1 mm, Geistlich Pharma AG, Wolhusen, Switzerland) overnight. Then, complexes were implanted subcutaneously into the backs of 6-week-old SCID beige mice (CB-17/Icr-scid-bg, male, the Vital River Laboratory Animal Technology Co., Beijing, China). The mice were randomly allocated to cages, with 4–5 animals housed per standard cage at 25 °C, and provided unlimited rodent chow and water. The NC group and FD group were transplanted with NC and FD BMSCs, respectively. PTHrP (20 μg/100 g, 5 times a week) was given to mice transplanted with NC BMSCs via intraperitoneal injection for three weeks. Subsequently, 6-thioguanine (6-TG, 100 μg/100 g, 5 times a week) was given to mice transplanted with FD BMSCs via intraperitoneal injection for three weeks. PTHrP was dissolved in PBS, while 6-TG was dissolved in 2% DMSO + 40% PEG 300 + 2% Tween 80 + ddH2Os. Samples were harvested eight weeks after implantation, and osteogenic levels were valued by H&E, Masson trichrome staining, and immunohistochemistry staining for OCN.

### 4.4. In Vitro Induction of Osteogenic Differentiation and Alizarin Red Staining

A total of 2 × 10^5^ BMSCs were plated onto six-well plates and cultured in osteogenic differentiation induction media containing α-MEM, 10% FBS, 50 μM ascorbic acid (Sigma-Aldrich, MO, USA), 100 nM dexamethasone (Sigma-Aldrich, MO, USA), and 10 mM β-glycerophosphate disodium salt hydrate (Sigma-Aldrich, MO, USA). The medium was replaced every three days. Alizarin red staining was performed after 14 or 21 days to evaluate the mineralization potential of BMSCs. Cells were fixed with 4% paraformaldehyde for 20 min and rinsed with double-distilled H_2_O. Subsequently, cells were stained with 1% alizarin red S solution (Solarbio, Beijing, China) for 10–15 min and rinsed with double-distilled H_2_O. Details of biological modulator information are provided in Appendix A.

### 4.5. Histological, Immunohistochemical, and Immunofluorescence Staining Assays

After fixation, decalcification, and paraffin embedding, 4 μm sections were prepared. Following standard blocking and antigen recovery processes, slides were incubated with primary antibodies overnight at 4 °C. The antibody information for IHC is shown in Appendix A. The following day, slides were incubated with corresponding secondary antibodies and stained using a DAB kit (Zhongshan Biosciences, Beijing, China).

Approximately 5 × 10^4^ cells were planted onto 24-well plates. After treatment with the appropriate medication for the specified duration, cells were fixed with 4% paraformaldehyde for 20 min. Then, cells were treated with 0.5% TritonX-100 (Solarbio, Beijing, China) and blocked by 3% BSA solution. Subsequently, cells were incubated with primary antibodies overnight at 4 °C. Fluorescence secondary antibodies and DAPI (Zhongshan Biosciences, Beijing, China) were utilized to accomplish immunofluorescence staining.

### 4.6. ELISA

Cell culture supernatants were collected and centrifuged for 20 min at 2000× *g* and 4 °C to remove insoluble components. The supernatants were collected for assays or stored at −80 °C for later use. Secreted PTHrP was tested by FineTest EH1058 (FineTest, Wuhan, China).

For cAMP measurement, cells in six-well plates were treated with a serum-free medium containing 0.5 mM 3-isobutyl-1-methylxanthine (IBMX, Selleck, TX, USA) for 1 h. Subsequently, cells were incubated with 0.1 M HCl for 20 min, and the lysate was immediately centrifuged at 2000× *g* for 10 min at 4 °C. The supernatants were collected to analyze the cAMP levels using an Enzo Direct cAMP ELISA kit (ADI-900-066, Enzo Life Sciences, NY, USA) or stored at −80 °C for later use.

### 4.7. Western Blot

The total protein content of the cells was extracted using a RIPA buffer (Solarbio, Beijing, China) with protease inhibitors and phosphatase inhibitors (Huaxingbio, Beijing, China). Proteins were loaded onto 10% or 12% SDS-PAGE and transferred to PVDF membranes (Millipore, MA, USA). Subsequently, the membranes were blocked with 5% non-fat milk for 2  h at room temperature, followed by incubation with primary antibodies overnight at 4 °C. The antibody information for the WB is shown in Appendix A. The membranes were stained with appropriate secondary antibodies the next day. To detect different antigens within the same blot, PVDF membranes were stripped using a Western Blot Stripping buffer (Huaxingbio, Beijing, China) and re-probed. Immunoreactive bands were visualized using an enhanced chemiluminescent detection reagent (NCM Biotech, Suzhou, China) and quantified by ImageJ 1.51j8 software.

For the drug concentration gradient experiments, the expression of p-CREB was normalized to CREB, while p-GSK-3β expression was normalized to GSK-3β, and p-PKA expression was normalized to PKA. The remaining gray values were normalized to GAPDH. Then, the mean gray value of the blank control group was calculated, and each gray value of the blank control group and experimental group was divided by the mean gray value of the blank control group to gain the relative protein quantification values. The relative gray value are displayed beneath the corresponding Western blot results.

### 4.8. ChIP and ChIP-qPCR

Chromatin immunoprecipitation (ChIP) assays were performed using a SimpleChIP^®^ Plus Enzymatic Chromatin IP kit (CST, MA, USA), following the manufacturer’s guidelines. Briefly, cells were cross-linked with 37% fresh formaldehyde and lysed using a lysis buffer. Subsequently, the cells were treated with micrococcal nuclease (MNase), sonicated briefly, and centrifuged at 9400× *g* to remove insoluble materials. Subsequently, the supernatants were collected and incubated overnight with anti-CREB antibodies (1:50, CST, USA, #9197) at 4 °C. After 24 h, the solutions were incubated with protein G magnetic beads at 4 °C for 2 h. Then, we washed the beads and reversed the cross-links of protein/DNA complexes to free the DNA. Real-time PCR (qPCR) was performed by ABI Prism 7500 (Applied Bioscience, PerkinElmer, Foster City, CA) after DNA was purified by spin columns. Primer sequences are shown in Appendix A.

### 4.9. Dual Luciferase Reporter Assay

The putative binding region of CREB in the human *PTHLH* promoter was amplified through PCR using genomic DNA and was cloned downstream of the firefly luciferase (FL) gene in the pGL3-basic luciferase reporter vector (Genomeditech, Shanghai, China). Co-transfected with the overexpression control (OC) vector and overexpression CREB vector, respectively, 293T cells were transfected with an empty vector (EV), a WT vector, two mutational binding site (MT1 and MT2) vectors, and a Renilla luciferase (RL) vector by a jetPRIME reagent (Polyplus transfection, Strasbourg, France). Co-transfected with the Renilla luciferase vector, FD BMSCs were transfected with an empty vector (EV), a WT vector, an MT1 vector, and an MT2 vector, respectively. Then, 48 h after transfection, cell lysates were collected and assayed by a Dual-Luciferase Assay kit (Beyotime, Shanghai, China) following the manufacturer’s instructions. Relative luciferase activity was calculated as the ratio of FL to RL activity.

### 4.10. siRNA Interference Assays

When cells reached 80–90% confluence, siRNA-NC or siRNA-*PTHLH* (RiboBio, Guangzhou, China) were transfected using the jetPRIME reagent. Then, 20 μM of siRNA and 5 μL of the jetPRIME transfection reagent were mixed in the transfection buffer. The complex was incubated at room temperature for 10 min and added to a complete medium. The silencing efficiency was detected by real-time PCR, WB, and ELISA assays.

### 4.11. Statistical Analysis

Statistical analysis and plots were performed using GraphPad Prism 7 software. All data examined are expressed as means ± SD. The in vitro experiments were performed with three biological replicates and at least three technical replicates. The statistical significance between the two groups was calculated by a Student’s *t*-test. A *p*-value < 0.05 was considered statistically significant and denoted by * or #; a *p*-value < 0.01 was denoted by ** or ##; a *p*-value < 0.001 was denoted by *** or ###; a *p*-value < 0.0001 was denoted by **** or ####.

## 5. Conclusions

In summary, this study demonstrates, for the first time, that PTHrP is a significant endogenous pathogenic factor in FD; it participates in initiating the cAMP/PKA/CREB signaling pathway and promotes the overactive Wnt/β-Catenin signaling pathway, leading to abnormal osteogenesis and proliferation in FD BMSCs. Furthermore, the positive feedback relationship between PTHrP and the cAMP/PKA/CREB signaling pathway might help elucidate the source of excessive PTHrP in FD lesions. Moreover, employing a small molecule inhibitor targeting PTHrP suggests an alternative therapeutic schedule to the FD phenotype by interfering with cell proliferation and osteogenic differentiation. This study provides novel insights into the possible pathogenesis involved in the FD phenotype and enhances our understanding of the molecular signaling pathways associated with FD.

## Figures and Tables

**Figure 1 ijms-24-07616-f001:**
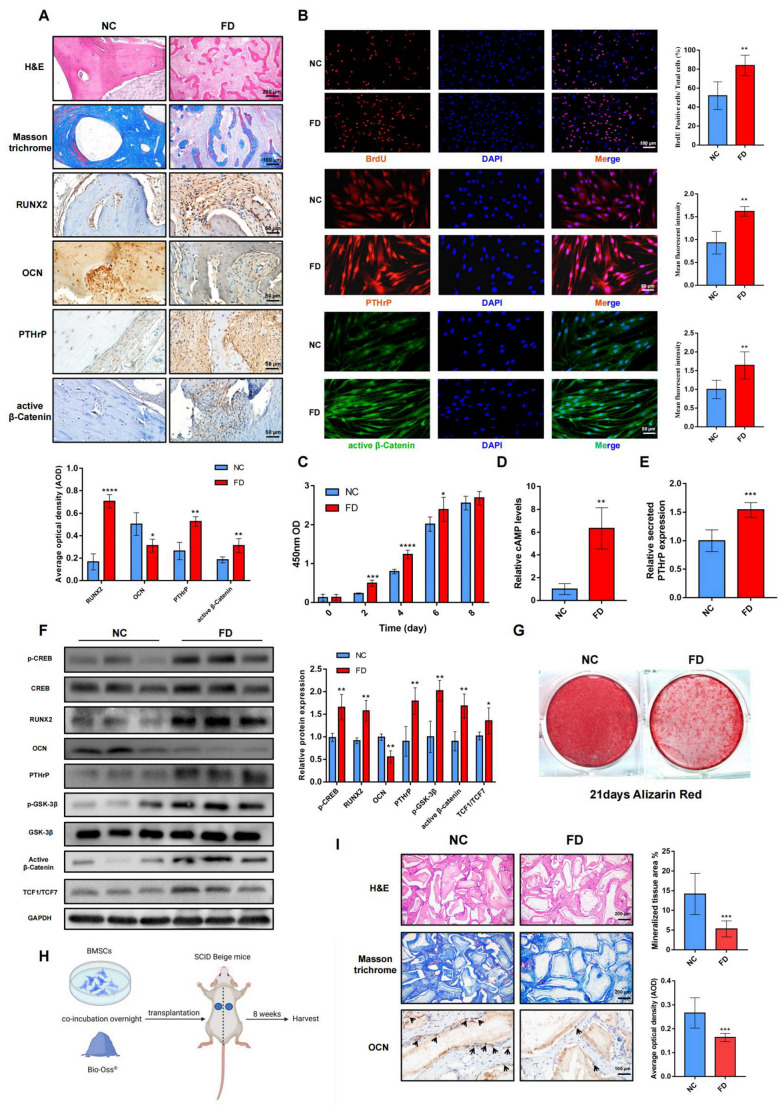
Biological characteristics of FD BMSCs. (**A**): H&E staining, Masson trichrome staining, and immunohistochemistry staining of RUNX2, OCN, PTHrP, and active β-Catenin in normal control patients and FD lesions. (**B**): Immunofluorescence staining of BrdU, PTHrP, and active β-Catenin in NC and FD BMSCs. (**C**): Cell proliferation tested by a CCK8 assay. (**D**): Relative cAMP levels in cells after 14 days of osteogenic differentiation. (**E**): Relative secreted PTHrP expression levels of NC BMSCs and FD BMSCs cultured in an osteogenic induction medium for 14 days. (**F**): Western blot analysis of osteogenesis markers of BMSCs after 14 days of osteogenic induction. For p-CREB expression, the relative protein level was normalized to CREB. For p-GSK-3β expression, the relative protein level was normalized to GSK-3β. The other gray values were normalized to GAPDH. (**G**): Alizarin red staining after 21 days of osteogenic differentiation. (**H**): BMSCs were loaded onto Bio-Oss^®^ and transplanted into both sides of the SCID beige mice’s backs. (**I**): H&E staining and immunohistochemistry of OCN in vivo. Black arrows: OCN positive osteoblasts. Bars indicate means ± SD. *p* > 0.05; *p* < 0.05, *; *p* < 0.01, **; *p* < 0.001, ***; *p* < 0.0001, ****.

**Figure 2 ijms-24-07616-f002:**
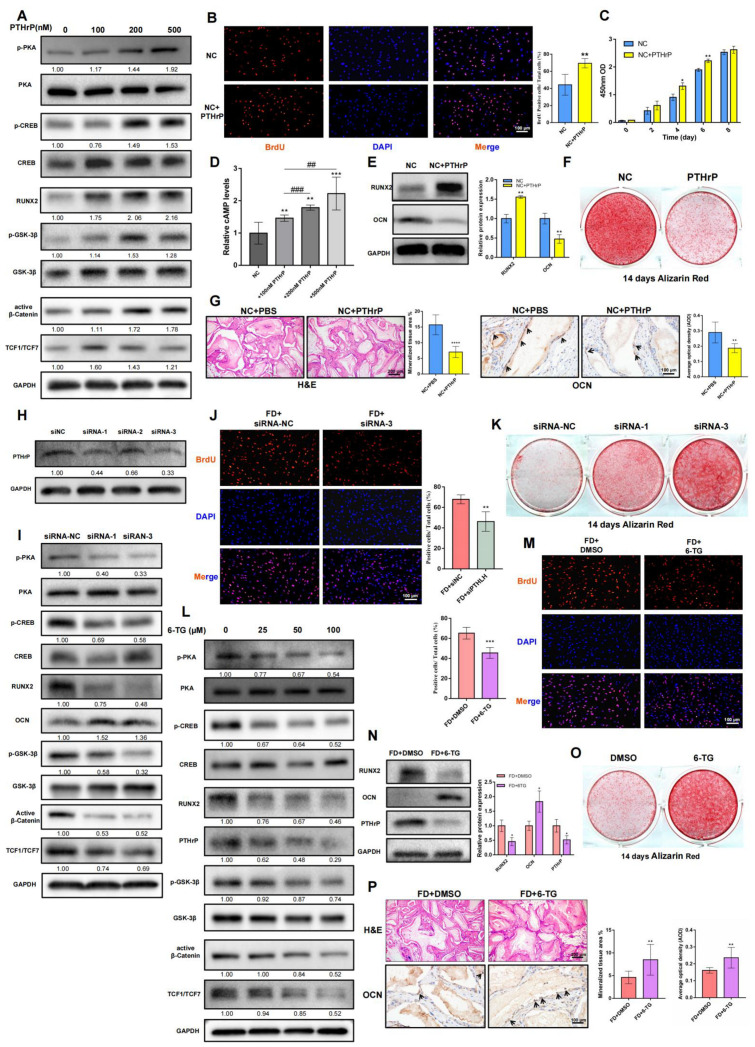
PTHrP promotes cell proliferation and inhibits osteogenesis. (**A**): WB analysis of NC BMSCs treated with PTHrP for 48 h at concentrations ranging from 0 to 500 nM concentration. Values represent the relative quantification of proteins at the indicated concentration. (**B**): Immunofluorescence staining of BrdU in NC BMSCs treated with PTHrP (200 nM) or PBS. (**C**): Proliferative capacity of NC BMSCs treated with PTHrP (200 nM) or PBS assessed by a CCK8 assay. (**D**): Trends in intracellular cAMP levels when treated with gradient PTHrP dosages. (**E**): Western blot analysis of RUNX2 and OCN in PTHrP (200 nM)-treated NC BMSCs cultured with an osteogenic induction medium for 14 days. (**F**): Alizarin red staining of cells cultured in an osteogenic medium for 14 days with or without PTHrP (200 nM). (**G**): H&E staining and immunohistochemistry staining of OCN in vivo. Black arrows: OCN positive osteoblasts. (**H**): Interference efficiency of three siRNA sequences detected by WB. (**I**): *PTHLH* transcription was knocked down by two independent siRNA. Protein expression levels of FD BMSCs after 14 days in an osteogenic medium culture were detected by WB. (**J**): Immunofluorescence staining changes of BrdU in FD BMSCs when treated with si-*PTHLH*-RNA3. (**K**): Alizarin red staining of FD BMSCs cultured in an osteogenic induction medium for 14 days with siRNA-NC, siRAN-1, and siRAN-3. (**L**): WB results of gradient 6-TG on FD BMSCs. (**M**): Immunofluorescence staining of BrdU in FD BMSCs when treated with 6-TG (50 µM) or DMSO. (**N**): Western blot analysis of RUNX2 and OCN in 6-TG (50 µM)-treated FD BMSCs cultured with an osteogenic induction medium for 14 days. (**O**): Alizarin red staining of FD BMSCs treated with 6-TG (50 µM) or DMSO after 14 days of osteogenic induction. (**P**): H&E staining and immunohistochemistry staining of OCN in vivo. Black arrows: OCN positive osteoblasts. Bars indicate means ± SD. *p* > 0.05; *p* < 0.05, *; *p* < 0.01, **/##; *p* < 0.001, ***/###; *p* < 0.0001, ****.

**Figure 3 ijms-24-07616-f003:**
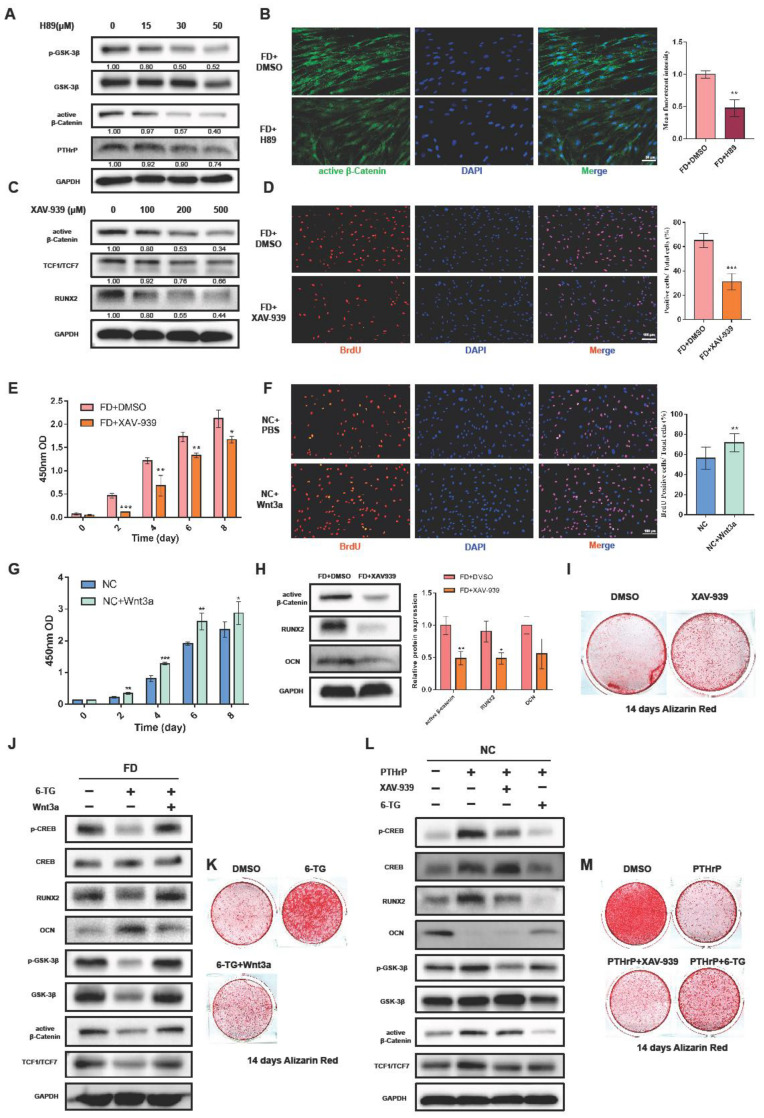
PTHrP could influence osteogenesis and proliferation of FD BMSCs through the canonical Wnt/β-catenin signaling pathway. (**A**): Treatment with varying concentrations of H89 for 48 h and WB analysis of active β-Catenin, p-GSK-3β, and PTHrP expression in FD BMSCs. (**B**): Immunofluorescence staining changes of active β-Catenin in H89 (30 μM)-treated FD BMSCs. (**C**): Effects of different concentrations of XAV-939 on FD BMSCs assessed by WB. (**D**): Immunofluorescence staining of BrdU, demonstrating proliferation changes in FD BMSCs after treatment with 200 μM XAV-939. (**E**): CCK8 results for FD BMSCs treated with XAV-939 (200 μM) or DMSO. (**F**): Immunofluorescence staining of the BrdU of Wnt3a (100 ng/mL)-treated NC BMSCs. (**G**): CCK8 results for NC BMSCs treated with Wnt3a (100 ng/mL) or PBS. (**H**): WB results of active β-Catenin, RUNX2, and OCN of XAV-939 (200 μM)-treated FD BMSCs cultured with an osteogenic induction medium for 14 days. (**I**): Alizarin red staining of FD BMSCs treated with XAV-939 (200 µM) or DMSO after 14 days of osteogenic induction. (**J**): WB analysis of Wnt3a (100 ng/mL) and/or 6-TG (50 µM)-treated FD BMSCs. (**K**): Alizarin red staining of FD BMSCs treated with Wnt3a (100 ng/mL) and/or 6-TG (50 µM) after 14 days of osteogenic induction. (**L**): WB analysis of XAV-939 (200 µM) or 6-TG (50 µM) on PTHrP (200 nM)-treated NC BMSCs. (**M**): Alizarin red staining of PTHrP (200 nM)-treated NC BMSCs with XAV-939 (200 µM) or 6-TG (50 µM) after 14 days of osteogenic induction. Bars indicate means ± SD. *p* > 0.05; *p* < 0.05, *; *p* < 0.01, **; *p* < 0.001, ***.

**Figure 4 ijms-24-07616-f004:**
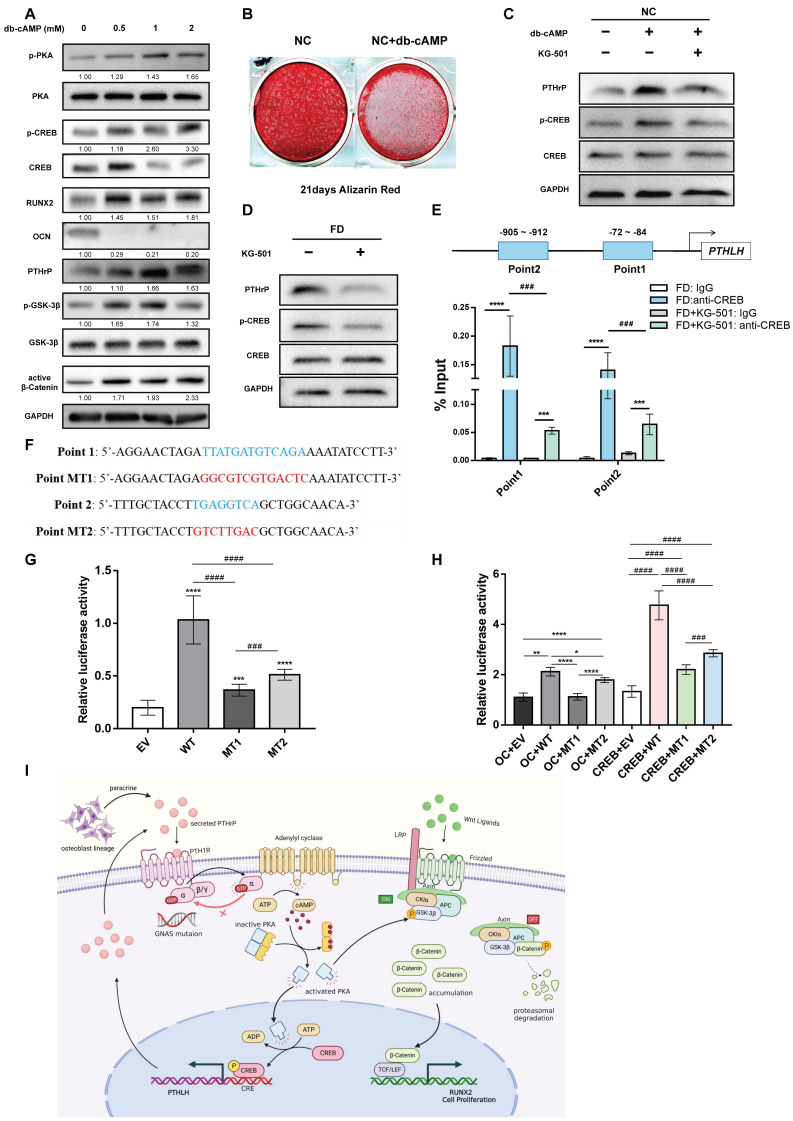
A positive feedback loop exists between PTHrP and the cAMP/PKA/CREB signaling pathway. (**A**): WB analysis of NC BMSCs treated with db-cAMP for 48 h at concentrations ranging from 0 to 2 mM. (**B**): Alizarin red staining of db-cAMP (1 mM)-treated NC BMSCs following 21 days of osteogenic induction. (**C**): WB analysis of inhibitory effects of KG-501 (25 μM) on p-CREB, CREB, and PTHrP in db-cAMP (1 mM)-treated NC BMSCs for 24 h. (**D**): WB analysis of the inhibitory effects of KG-501 (25 μM) on p-CREB, CREB, and PTHrP in FD BMSCs for 24 h. (**E**): Two predicted binding sites between CREB and the *PTHLH* promoter. ChIP-qPCR results of two putative CREB binding sites in the *PTHLH* promoter region in FD BMSCs with or without KG-501 (25 μM). (**F**): Sequences of two predicted binding sites (blue words) and corresponding mutation sites (red words). (**G**): Relative luciferase activity of FD BMSCs transfected with an empty vector (EV), a wild-type *PTHLH* promoter luciferase reporter (WT), a mutational point 1 promoter luciferase reporter (MT1), and a mutational point 2 promoter luciferase reporter (MT2). (**H**): Relative luciferase activity of 293T cells transfected with the overexpression control vector (OC) or overexpression CREB vector. (**I**): The effect of PTHrP on cAMP/PKA/CREB and Wnt/β-catenin signaling pathways in FD BMSCs. Bars indicate means ± SD. *p* > 0.05; *p* < 0.05, *; *p* < 0.01, **; *p* < 0.001, ***/###; *p* < 0.0001, ****/####.

## Data Availability

Data are available on request from the authors.

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
