# Peer review of "PTHrP Modulates the Proliferation and Osteogenic Differentiation of Craniofacial Fibrous Dysplasia-Derived BMSCs"

_ijms, 2023, doi:10.3390/ijms24087616_

Round 1

Reviewer 1 Report

Dear  Authors, this work is really well performed and clearly shows the link between PTHrP and craniofacial fibrous dysplasia pathogenesis. Furthermore, it provides possible treatment or illness management strategies. However, there are some mistakes and inaccuracies, which should be improved before publishing:

1) There are a number of spelling mistakes in the text, e.g. in line 62 'it' is written in upper case letter, in line 159 'knock down' should be written as separate words. Please spell-check it one more time.

2) In figures 2A, 2I, 2L, and others is not clear how and for which proteins the rations were calculated. Please calculate for all the proteins, or better explain it in the text or figures caption.

3) Line 139: Sustained administration of exogenous PTHrP to NC BMSCs upregulated expression of p-PKA, p-CREB and cAMP obviously, representing the activation of 140 cAMP/PKA/CREB signaling pathway (Fig. 2A and 2D). Authors should not mix expression and phosphorylation terminology. When talking about total forms of the protein use term expression, when talking about phosphorylated proteins forms, talk about phosphorylation levels. Furthermore, cAMP is not a protein, it is not expressed it is produced or synthesised in the cell. Do not mix this terminology. It is mentioned in more than one place in the text.

3) Fig 2D, Fig S3 E-g - correct y axis. c-AMP levels.

4) Line 153 The same wrong usage of terminology. First proteins are expressed, then they are modified.

5) Please provide x axis labels in proliferation results.

6) In line 458 there is a missing concentration of beta-glycerolphosphate, please provide it.

7) Line 469, provide supplemenent table number

8) Line 486, again, wron usage of the terminology.

Reviewer 2 Report

Minor issues:

1.       Your details on patient information needs to be further enhanced. Please write a paragraph clearly in the materials and methods section outlining the age range of patients, total ‘n’ number of patients, male to female ratios. The same details need to be added for the controls as well.

2.       Please provide details of the mice strain used for the in vivo experiments

3.       For your data being presented as mean – was it examined for normality? Usually, working with human donor samples, the data is no normally distributed. In such situations, median is a better choice for data representation. Please perform the normality test for all your data sets. If the data set is found to be normal – mean is fine. If it’s not – please rearrange and represent the data with median values.

4.       Lines 534 and 535 – ‘p’ should be in small letters and not I capitals for indicating significant values. Please apply this for all your figure legends as well.

5.       Majority of your bar graphs have error bars indicating that you have performed your experiments in duplicates/triplicates. However, that is not clear from your methods. Have you performed everything in biological replicates only (n=3 patients and n=3 control) or have you also performed technical replicates? It is currently unclear from your article.

6.       Please add a paragraph on limitations of the study and future directions. Please reflect on your work and suggest things that have acted as a limitation to this study (for example only n=3 donor samples, any other experiments that could have also been performed etc). For future direction, please iterate how you think this work may be taken forward for potential use in humans and what would be the next steps. This will ensure our readers that you have a vision of your work.

Grammatical/editing suggestions:

1.       Please ensure that the font for the contact emails is same as the rest of the text.

2.       Please mention the full-form of GNAS and cAMP in the first line of the abstract.

3.       The font of the references is different from that of the text. Please unify that.

4.       Please proof read for grammatical errors – needs to be read by someone who is an English language expert.

5.       Ideally, all the names of different genes should be in italics.

Needs moderate grammatical editing
